# Vitamin A Concentration in Human Milk: A Meta-Analysis

**DOI:** 10.3390/nu14224844

**Published:** 2022-11-16

**Authors:** Huanmei Zhang, Xiangnan Ren, Zhenyu Yang, Jianqiang Lai

**Affiliations:** 1Department of Maternal and Child Nutrition, National Institute for Nutrition and Health, Chinese Center for Disease Control and Prevention, Beijing 100050, China; 2China-DRIs Expert Committee on Human Milk Composition, Chinese Nutrition Society, Beijing 100050, China; 3National Institute for Nutrition and Health, Chinese Center for Disease Control and Prevention, Beijing 100050, China

**Keywords:** vitamin A, retinol, human milk, full-term infant, lactation stage

## Abstract

Humans require vitamin A (VA). However, pooled VA data in human milk is uncommon internationally and offers little support for dietary reference intake (DRIs) revision of infants under 6 months. As a result, we conducted a literature review and a meta-analysis to study VA concentration in breast milk throughout lactation across seven databases by August 2021. Observational or intervention studies involving nursing mothers between the ages of 18 and 45, with no recognized health concerns and who had full-term infants under 48 months were included. Studies in which retinol concentration was expressed as a mass concentration on a volume basis and determined using high-, ultra-, or ultra-fast performance liquid chromatography (HPLC, UPLC, or UFLC) were chosen. Finally, 76 papers involving 9171 samples published between 1985 and 2021 qualified for quantitative synthesis. Results from the random-effects model showed that the VA concentration of healthy term human milk decreased significantly as lactation progressed. VA (µg/L) with 95% CI at the colostrum, transitional, early mature and late mature stages being 920.7 (744.5, 1095.8), 523.7 (313.7, 733.6), 402.4 (342.5, 462.3) and 254.7 (223.7, 285.7), respectively (X^2^ = 71.36, *p* < 0.01). Subgroup analysis revealed no significant differences identified in VA concentration (µg/L) between Chinese and non-Chinese samples at each stage, being 1039.1 vs. 895.8 (*p* = 0.64), 505.7 vs. 542.2(*p* = 0.88), 408.4 vs. 401.2 (*p* = 0.92), 240.0 vs. 259.3 (*p* = 0.41). The findings have significant implications for the revision of DRIs for infants under six months.

## 1. Introduction

VA involves various physiological processes including retinal vision, gene expression, immune strength, reproduction, embryonic development, and growth [1,2]. Latest research shows that VA may have protective effects on outcomes of some viral infections such as HPV and measles [3]. Breastfed infants, particularly exclusively breastfed infants, acquire VA through human milk to achieve such needs. Breast-milk VA is a good indication of infants and lactating mothers’ VA status, according to the World Health Organization (WHO) [4,5]. The WHO, European Food Safety Authority (EFSA), and other competent scientific organizations have established dietary reference VA values for infants and lactating women [2,6,7]. However, such recommendations by the WHO or EFSA were based on hypotheses, carried out with limited research and few subjects [8]. There have been quite a few publications about human milk VA concentrations worldwide; however, there is a wide range as shown in publications [9,10,11]. Meta-analysis methodology has been viewed as a beneficial method for statistically combining and summarizing the results from various studies, so as to obtain pooled data or estimates that may better represent what is true in the population [12]. However, relatively few systematic reviews and meta-analyses have been undertaken to synthesize VA concentration in human milk using international data [13,14], resulting in scarcity of updated and solid breast milk VA levels. Such data is critical for establishing dietary reference intakes (DRIs) for groups, particularly newborns and breastfeeding mothers [8].

To inform DRI revision for the group of healthy full-term infants aged 0 to 6 months, we conducted a meta-analysis study to analyze human milk’s VA concentration on volume base, determined by advanced methods (HPLC, UPLC, or UFLC), and to explore the influence of potential confounders using meta-regression.

## 2. Materials and Methods

### 2.1. Literature Search

Articles in English were searched through PubMed, Web of Science, Embase and Cochrane Central Register of Controlled Trials employing matching keywords “vitamin A” or “retinol” with “human * milk”, “woman * milk”, “mother * milk”, “breast * milk”, “lactation” or “lactating”. Articles in Chinese were searched through the China National Knowledge Internet (CNKI), Wan Fang Database, or China Science and Technology Journal Database (CSTJ) utilizing matching Chinese keywords covering “human milk” and “vitamin A”, the former keyword including “human milk”, “breast milk”, or ”lactating mother” and the latter including “vitamins”, “vitamin A”, “retinol”, “nutrients” or “nutritional composition”. Articles published by 21 August 2021 were taken into account. 

### 2.2. Study Selection and Screening

Studies were considered if they matched the following criteria: (1) intervention or observational studies that reported the level of VA in human milk; (2) mothers were aged between 18 to 45 years; (3) healthy lactating mothers free of degenerative or metabolic illnesses; (4) full-term infants aged 0 to 48 months; and (5) high-, ultra-, or ultra-fast performance liquid chromatography (HPLC, UPLC or UFLC) determination method. Criteria for exclusion: (1) VA supplementation or particular diet intervention was used; (2) studies were presented as a review, case report, conference abstract, or proceedings without full-text articles, communication letters, texts described in language other than English or Chinese, duplicate publications, or full-text inaccessible; (3) preterm milk studies or data derived from a blend of preterm and term breast milk; (4) research with identical samples, or concentration data not in volume unit, inconsistent data, or unusual data; (5) no clear identification of lactation stage. EndNote version 20.3 (Clarivate Analytics (UK) Limited, London, UK) was used to screen and choose studies.

Data of Orhon et al. [15] at the colostrum stage, Vaisman et al. [16] at the transitional phase and Redeuil et al. [9] at the transitional and late mature stages were removed due to unusual data distribution. That of Eagle-Stone et al. [17] was omitted because participants in some regions likely took high-dose VA supplements three months before the survey. The supplementing effect on raised VA levels of human milk was considered durable within 6 months [4]. 

### 2.3. Data Extraction

Papers were chosen in the order of title, abstract and entire contents based on the inclusion and exclusion criteria stated above. Four detectives extracted and double-checked the data. A fifth investigator was consulted for assistance if there was any doubt throughout the selection process. First author, publication year, study design, study location, analysis methods, participant characteristics (sample size, mother age, lactation stage) and VA concentration of human milk sample were all extracted. 

Data from various lactating stages were retrieved and included in cohort studies. A median point was chosen if more than one human milk sample subgroup was tested within the same lactation stage. In intervention studies, baseline data from the intervention and control groups were retrieved, and follow-up data from the control group were treated similarly to cohort research. When both full breast-milk samples and random samples were used in the same study, data of the former was extracted. All extracted data was recorded using Excel. 

### 2.4. Data Analysis

The VA concentration data were given as mean± standard deviation of retinol. If retinyl palmitate levels were also given, they were converted in proportion. If provided as geometric mean and 95% CI, geometric mean was deemed identical to arithmetic mean values, and 95% CI was regarded arithmetic mean values. Zhang et al. [18] were consulted on the data transformation. The transformation of VA concentration in human milk from mass base to volume was multiplied by a factor of 1.032. Individual sample size formation was used in the computation when many samples were used in a single study, and weighted mean and standard deviation were required. The following are the relevant functions:Mean = (n1 × M1 + n2 × M2 + n3 × M3 + … + ni × Mi)/(n1 + n2 + n3+ … + ni)(1)
A_i_ = S_i_^2^ (n_i_ − 1) + M_i_^2^ × n_i_
(2)
(3)$$\mathrm{SD}=\sqrt{\frac{\sum A_{i}-\frac{{\left[\sum \left(M_{i}n_{i}\right)\right]}^{2}}{N}}{N-1}}$$

The individual mean value, sample size and standard deviation of personal research are represented by Mi, ni and Si, respectively.

A weighted mean and standard deviation were determined for a common lactation stage. If the VA result of an individual study exceeded the range of Mean ± 2SD, the data was considered an outlier and was excluded. The other studies were then incorporated and integrated using meta-analysis with a random-effects model because there are differences between studies in both population and performance. Subgroup analysis at each lactation stage was conducted by countries: Chinese and non-Chinese studies.

The I^2^ test with a significance level of α = 0.05 was used to visually analyze heterogeneity among studies regarding the human milk VA level and to quantify the magnitude of heterogeneity. Individual trials were examined using sensitivity analysis at each stage of breastfeeding. Sources of heterogeneity were assessed using meta-regression analysis, which included research design, publication year, mother’s age, country, sampling time, sampling volume, and whether the breast was empty after sampled. R packages version 4.1.3 (10 March 2022) were used for meta-analysis and meta-regression.

## 3. Results

### 3.1. Study Identification

Database searching yielded 12,887 entries, including 11,089 abstracts in English and 1798 in Chinese (Figure 1). After all duplicates were removed, 9558 records were tested against the title and abstract. In addition, three articles were included during the paper-chasing procedure [18,19,20]. To establish eligibility for inclusion in the review, we evaluated 118 full-text studies. Finally, 76 studies from 33 countries met the inclusion and exclusion criteria. They were assessed for data review, with seventy-one full-texts in English [9,10,11,16,20,21,22,23,24,25,26,27,28,29,30,31,32,33,34,35,36,37,38,39,40,41,42,43,44,45,46,47,48,49,50,51,52,53,54,55,56,57,58,59,60,61,62,63,64,65,66,67,68,69,70,71,72,73,74,75,76,77,78,79,80,81,82,83,84,85,86,87], four in Chinese [88,89,90,91], and one in Spanish but with abstract in English [74]. 

### 3.2. Study Characteristics

There was one human milk bank study, forty cross-sectional studies, sixteen randomized-control studies, four cohort studies, eleven intervention studies, one cross-sectional study in parallel to one intervention study, and three longitudinal studies (Table 1), with six studies involving Chinese participants, sixty-nine studies involving non-Chinese participants and one study involving multinational participants. A total of 9171 human milk samples were included, with VA concentrations in colostrum, transitional and mature human milk determined in 2170, 719 and 6282 models, respectively. There were 4082 and 950 samples included as early mature and late mature human milk, respectively, yet 999 specimens with no clear indication whether they were early or late mature. There were 2053 Chinese and 5602 non-Chinese participants. These studies were published between Year 1985–2021.

### 3.3. VA Concentration in Human Milk

The VA concentration (µg/L) with 95% CI of human milk at colostrum, transitional, early and late mature stages was 920.7 (744.5, 1096.8), 523.7 (313.7, 733.6), 402.4 (342.5, 462.3) and 254.7 (223.7, 285.7) for all samples, respectively (Table 2). The VA concentration with 95% CI of mature human milk was 385.3 (339.4, 431.3) µg/L. Subgroup analysis by lactation stage showed there were significant difference between the colostrum, transitional and mature stages (X^2^ = 170.02, *p* < 0.01) (Appendix A) and between the colostrum, transitional, early and late mature stages (X^2^ = 71.36, *p* < 0.01) (Appendix A).

At the colostrum stage, which is within 7 days following delivery, there were five studies performed on 429 Chinese participants and twenty-two studies completed on 1741 non-Chinese subjects (Figure 2). The VA content with 95% CI was 1039.1 (470.3, 1607.8) µg/L in Chinese specimens and 895.75 (714.1, 1077.4) in non-Chinese. There was no statistically significant variance between the two population groups (X^2^ = 0.22, *p* = 0.64). 

At the transitional stage, which is postpartum 8–14 days, there were three studies carried out among 356 Chinese subjects and three studies among 363 non-Chinese subjects (Figure 3). The VA concentration with 95% CI was 505.7 (118.0, 893.4) µg/L for Chinese and 542.2 (278.9, 805.6) for non-Chinese samples, respectively. There was no significant difference between the two populations (X^2^ = 0.02, *p* = 0.88). 

There were seven studies conducted among 1268 Chinese subjects and fifty-three studies among 5014 non-Chinese participants at the mature human milk stage (Appendix A), covering seven studies with 1112 Chinese and thirty-eight studies with 3221 non-Chinese at early mature stage (Figure 4), one study with 156 Chinese and five studies with 794 non-Chinese subjects at late mature stage (Figure 5). The VA concentration with 95% CI between Chinese and non-Chinese participants was 386.4 (270.6, 502.3) µg/L vs. 385.2 (335.1, 435.3) µg/L at the mature stage (X^2^ = 0.00, *p* = 0.98), 408.4 (282.6, 534.1) µg/L vs. 401.2 (333.6, 468.8) µg/L at the early mature stage (X^2^ = 0.01, *p* = 0.92) and 240.0 (214.9, 265.1) µg/L vs. 259.3 (220.8, 297.8) µg/L at the late mature stage (X^2^ = 0.68, *p* = 0.41). There was no significant difference when comparing population subgroups at the mature, early mature or late mature lactation stage. 

### 3.4. Heterogeneity and Sensitity Analysis

All analyses revealed substantial heterogeneity (I^2^ in 85~100%). Following sensitivity testing, no significant change in the combined effect of VA levels was seen at each lactation stage, indicating that all respective synthesized results was stable.

### 3.5. Meta-Regression

The results of the univariate meta-regression analysis revealed that none of the following, i.e., publication year, sampling time, whether emptying breast or not after sampling, whether Chinese or not, or study design type at each lactation stage, were significantly associated with heterogeneity between studies (all *p* > 0.05) except maternal age (≥30 years vs. <30 years) and nationality (Table A1). The explained heterogeneity of country changed very little following correction for maternal age, i.e., 54.58% to 54.20% at early mature human milk stage (both *p* < 0.0001), but the effect of maternal age changed from significant to insignificant, i.e., its *p* value being increased from 0.025 to 0.051. Equally, at the colostrum stage, the explained heterogeneity of country after correction resulted in a minute change, i.e., 21.56% to 25.94% but with the *p* value decreasing from 0.34 to 0.044, whereas the impact of maternal age changed from significant to insignificant again, with the *p* value increasing from 0.041 to 0.48. The results of the multivariate meta-regression study results suggested that country was a source of heterogeneity, while maternal age was not.

## 4. Discussion

In this study, we compiled previously published data on retinol concentrations in term human milk at each step of the four lactation stages and compared them between Chinese and non-Chinese studies. Our research comprised 76 articles, including 9171 participants from 33 countries from 1985 to 2021, for calculating human milk VA levels determined by HPLC, UPLC, or UFLC. At the colostrum, transitional, early mature and late mature stages of human milk, the VA levels were 920.7 µg/L (3.21 µmol/L), 523.7 µg/L (1.83 µmol/L), 402.4 µg/L (1.40 µmol/L), and 254.7 µg/L (0.89 µmol/L), respectively. There was no significant difference in the VA levels between Chinese and non-Chinese human milk at each lactation stage. This research has crucial implications for DRIs VA modification.

### 4.1. Data Interpretation

Our findings were compatible with previous meta-analysis findings on VA levels in human milk and the declining tendency with the lactation stage. Dror et al. [14] examined retinol levels in colostrum in four included studies and the mature stage in twenty-four studies (21~365-day lactation). The systematic approach chose the retinol-to-fat ratio (µmol/g fat) as the primary outcome measure which led to nearly two-thirds of the relevant literature being excluded for the meta-analysis. Despite this, the outcomes of our research at the two segmental stages were similar to those of Dror et al., namely 920.7 µg/L vs. 999.7 µg/L, 385.4 µg/L vs. 383.8 µg/L. de Vries et al. [13] conducted a systematic review of 11 studies on the relationship between colostrum VA and maternal serum (plasma) vitamin concentration but did not carry out a meta-analysis. As a result, our findings have greater precision and comprehensiveness.

The respective wide data distribution could explain the similar VA level of human milk at between Chinese and non-Chinese individuals at each lactation stage. Typically, the samples by Zhang et al. [91] were from 20 counties in 11 provinces across China, including urban and rural locations. In contrast, the non-Chinese samples came from 32 nations, comprising both developed, developing, and under-developed ones. Our multivariate meta-regression results, on the other hand, showed that the country factor explained more than 50% of the heterogeneity, implying that the remaining variation between studies could be due to factors such as VA intake, maternal status or sampling protocol rather than the insignificant factors such as study design, publication year, and so on that we analyzed here. Previous research has shown that inadequate dietary VA consumption, maternal VA status during pregnancy and lactation all contribute to clinical heterogeneity, while breast milk sampling protocol accounts for methodological heterogeneity [53,63,66,70,92].

### 4.2. Implications of Our Results for DRIs Revision

Two studies reported mean liver VA concentration of perinatal normal-weight newborns [93,94], one being 17.3 ± 17.4 µg/g liver in Thai fetuses in gestational age of 37–40 week (*n* = 10), the other being 22 ± 26 µg/g liver in USA infants aged 0–6 days (*n* = 22). Assuming that the liver represents 4.3% of body weight and the liver VA concentration is 20 µg/g, a 3.2-kg full-term newborn has stores of 2.8 mg VA. In contrast, an exclusively breast-fed infant consumes approximately 54.3 mg of VA from mother’s milk (402.4 µg/L × 0.75 L/day × 180 days). About 19.4 times more VA is transferred from a mother to a baby during the 6 months of lactation than is accumulated by the fetus during 9 months of gestation. Obviously, the VA in breast milk is of paramount importance for maintaining adequate VA status in early postnatal life of infants as compared to accumulation of VA in the liver prenatally.

A proper estimation of human milk VA level is critical for reference setting in terms of dietary adequate intake requirement for the population of exclusively breastfed infants under six months of age to guarantee optimal growth and development of the newborns. Accurate VA adequacy information for newborns and nursing mothers is desperately needed [8]. As a result, we proposed that the VA content in early mature human milk expressed as a mean with a 95% CI of 402.4 (342.5, 462.3) µg/L or 1.40 (1.20, 1.61) µmol/L, could be used as data support for the purpose. First, a VA level in human milk greater than >1.05 µmol/L may prevent clinical VA deficiency during the first six months of infancy [5]. Second, in this investigation, the synthesis result of VA level at the early mature stage showed less variance than colostrum and transitional phase and had a greater level than at later mature stage, indicating a better representative of human milk VA level. Third, an equilibrium of VA secretion appeared to be obtained in early mature human milk for human beings, as evidenced by the relatively comparable levels of VA in both Chinese and non-Chinese participants at this stage. It is worth noting that 923 mother-infant dyad participants in the Zhang 2021 [91] study had generally adequate nutrition and health status. The comparable VA concentration in early mature human milk is 0.25 mg/L (0.87 µmol/L). This threshold is far lower than the level proposed here. Fourth, past values presented by authoritative groups might have been overstated. The current acceptable intake level of VA for infants aged 0 to 6 months established by EFSA or IOM was based on the average amount of VA consumed in humans [1,2]. However, if the figures are derived from a small number of articles, there is a risk of overestimating the average demand for this group. The EFSA limit of 530 μg/L, chosen as the midpoint of a range of averages (229–831) μg/L, was based on five studies conducted in western countries that did not differentiate between early mature and later-stage human milk. Based on four investigations [19,20,21,32], the IOM established 485 μg/L (1.70 µmol/L) as the VA level in human milk in 2001 and adopted the level from one of the studies, which was undertaken among three healthy, well-nourished mothers within 75~277 days postpartum [21]. Our data analysis included these four research studies, whereas one study [19] was omitted due to outdated methodology. According to EFSA [7], the average levels of total VA concentration in western countries have generally been estimated to be between 450 and 600 μg/L. Nevertheless, we proposed a reference concentration range of VA in human milk of 402.4 (95% CI: 342.5, 462.3) μg/L for DRIs VA modification for infants ≤ 6 months of age. The range could also be helpful to nursing women and to optimizing the VA level of infant formula.

### 4.3. Study Limitations

Since the availability of the individual studies limited our meta-analysis of studies at the transitional and later mature lactation stages, the influence of country variability on human milk VA level at transitional stage may be weak, and it was not possible to assess it at the later mature stage. As a result, the meta-regression results for these two stages should be interpreted with caution.

## 5. Conclusions

The current study found that synthesized human milk VA levels decreased as breastfeeding progressed and that there was no significant difference in human milk VA levels between China and other countries, even though country played a vital role in the variation. Our findings have important implications for DRIs VA revision for the population of exclusively breastfed infants under six months.

## Figures and Tables

**Figure 1 nutrients-14-04844-f001:**
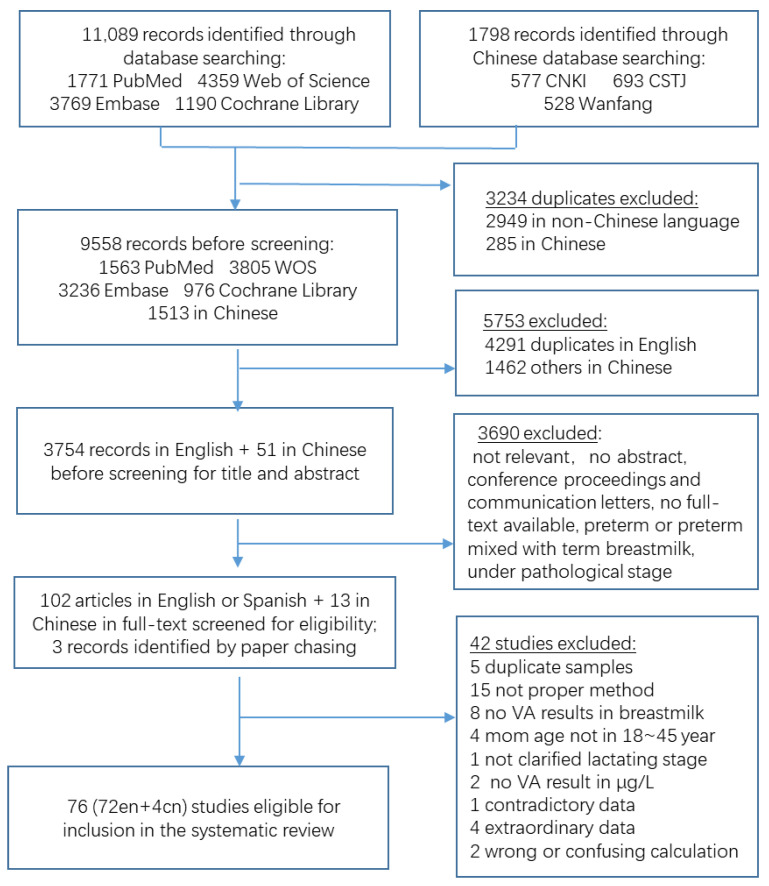
Flow diagram of literature review.

**Figure 2 nutrients-14-04844-f002:**
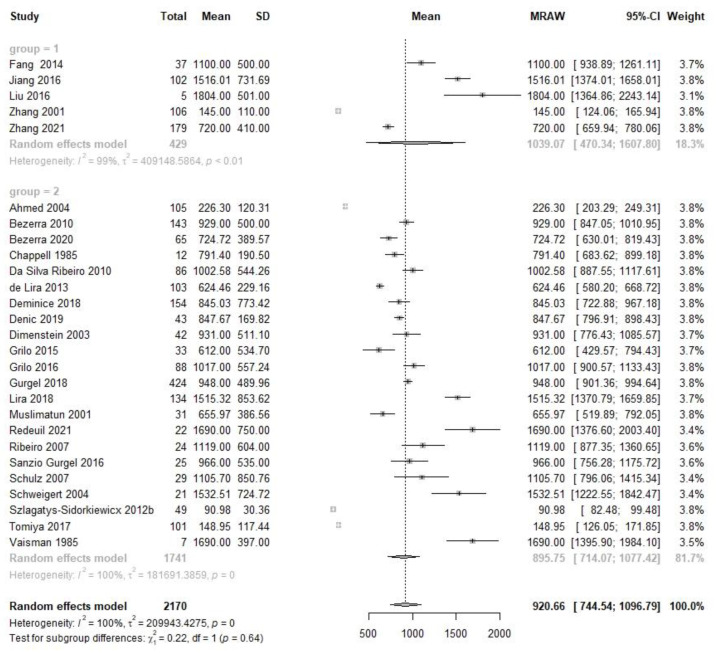
Forest plot of colostrum VA concentration and subgroup analysis between Chinese and non-Chinese samples.

**Figure 3 nutrients-14-04844-f003:**
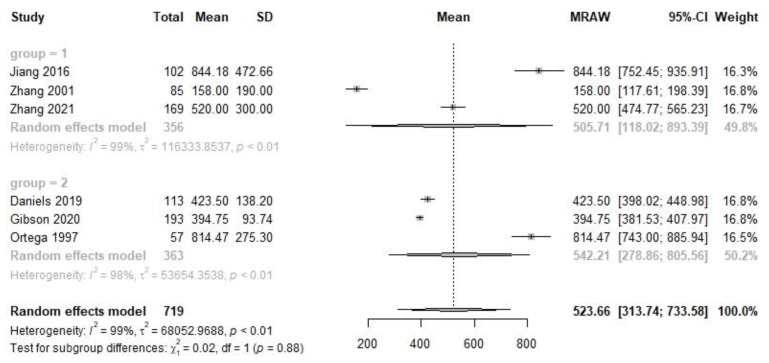
Forest plot of transitional human milk VA concentration and subgroup analysis between Chinese and non-Chinese samples.

**Figure 4 nutrients-14-04844-f004:**
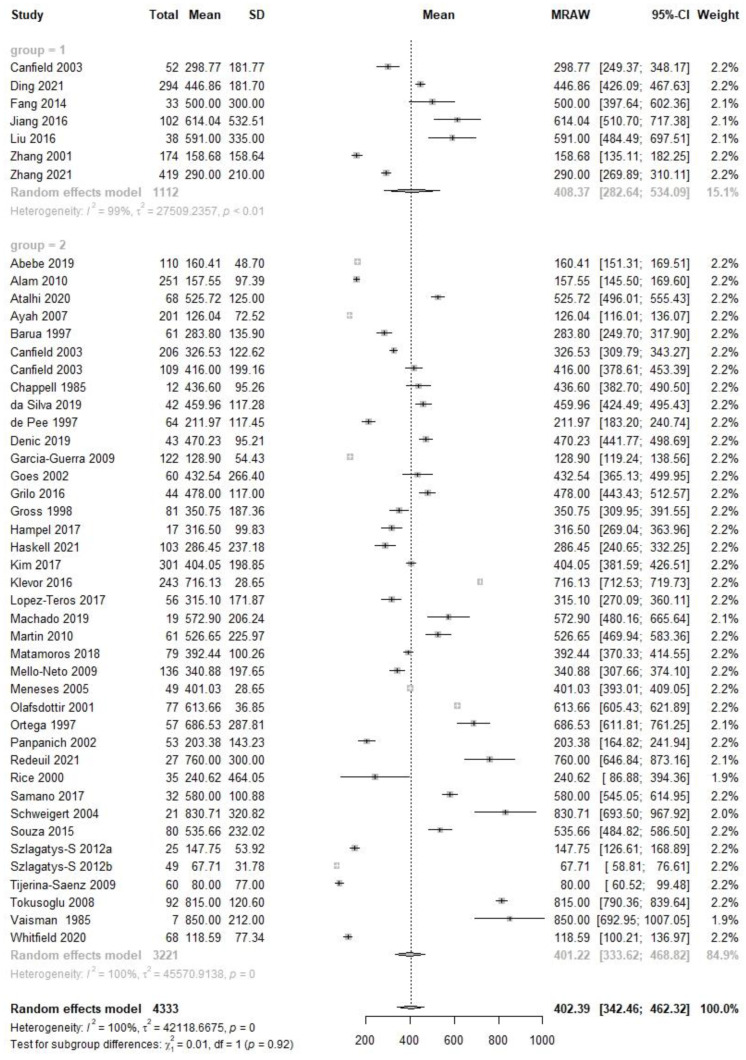
Forest plot of early mature human milk VA concentration and subgroup analysis between Chinese and non-Chinese samples.

**Figure 5 nutrients-14-04844-f005:**
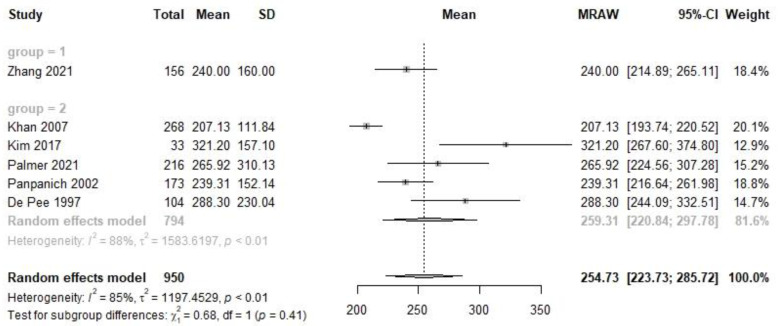
Forest plot of late mature human milk VA concentration and subgroup analysis between Chinese and non-Chinese samples.

**Table 1 nutrients-14-04844-t001:** Summary Characteristics of included studies.

First Author and Publication Year	Country	Study Design	Postpartum Days	Lactation Stage	Age of Mothers (y)	Subjects	Sample Size	Empty Breast or Not	Sampling Time ^#^
Abebe 2019 [22]	Ethiopia	CSS	180	3	23.0~33.6	110	110	No	AM
Agne-Djigo 2012 [23]	Senegalese	IS	156–198	3 + 4	20.9~35.3	59	59	Yes	AM
Ahmed 2004 [10]	Bangladesh	CSS	2	1	18.7~27.5	105	105	No	NS
Alam 2010 [24]	Bangladesh	RCT	61–89	3	21~33	251	251	No	AM + PM
Atalhi 2020 [25]	Morocco	IS	15	3	19~40	68	68	No	AM
Ayah 2007 [26]	Kenya	RCT	92–98	3	18.0~30.8	201	201	NS	NS
Barua1997 [27]	Bangladesh	CSS	45–780	3	18–32	61	61	No	AM
Bezerra 2010 [11]	Brazil	IS	1	1	19.2~29.8	143	143	No	AM
Bezerra 2020 [28]	Brazil	CSS	1	1	19.4~30.2	65	65	No	AM
Canfield 1997 [21]	China	IS	75–277	3 + 4 *	24.8~32.6	6	6	Yes	PM
Canfield 1998 [20]	China	IS	30~298	3 + 4	20.8~35.6	3	3	Yes	PM
Canfield 1999 [31]	Honduras	IS	30~365	3 + 4	17.3~30.1	36	36	Yes	AM
Canfield 2001 [30]	Honduras	IS	90–330	3 + 4	19.5~32.5	79	79	no	AM
Canfield 2003 [29]	Multination ^&^	CSS	25–193	3, 3 + 4	24.6~30.4	471	471	Yes	PM
Chappell 1985 [32]	Canada	CS	1~25	1, 3	NA	12	24	Yes	AM
da Silva 2019 [33]	Brazil	CS	25–134	3	20.4~35.2	42	42	No	NS
da Silva 2010 [34]	Brazil	CSS	1	1	19.6~31.2	86	86	Yes	AM
Daniels 2019 [35]	Indonesia	CSS	14	2	19.7~31.9	113	113	Yes	AM
de Lira 2013 [36]	Brazil	CSS	1–3	1	17~31	103	103	no	AM
de Pee 1995 [37]	Indonesia	RCT	150–384	3 + 4	17~40	175	175	yes	AM
de Pee 1997 [38]	Indonesia	CSS+ IS	90–180, 181~548	3, 4	17~40	168	168	Yes	AM
Deminice 2018 [39]	Brazil	CSS	2~6	1	20.3~31.4	154	154	No	NS
Denic 2019 [40]	Serbia	CSS	1~30	1, 3	18~40	43	86	Yes	AM
Dimenstein 2003 [41]	Brazil	CSS	1~2	1	18~39	42	42	No	AM + PM
Ding 2021 [42]	China	RCT	30–45	3	26.0–34.9	294	294	No	AM
Duan 2021 [43]	South Korea	CSS	NA	3 + 4	NA	34	34	NS	NS
Duda 2009 [44]	Poland	CSS	30–360	3 + 4	25.7~31.7	30	30	NS	NS
Ettyang 2004 [45]	Kenya	CSS	14~450	3 + 4	23~35	62	62	No	random
Fang 2014 [88]	China	CSS	3–30	1, 3	NA	70	70	NS	NS
Garcia-Guerra 2009 [46]	Mexico	IS	30	3	18~28.8	122	122	Yes	AM + PM
Gibson 2020 [47]	Indonesia	LS	60–150	2	22~34.8	193	193	Yes	AM
Goes 2002 [48]	Brazil	CSS	30–180	3	NA	60	60	NS	NS
Grilo 2015 [49]	Brazil	RCT	1	1	18–35	33	33	No	AM
Grilo 2016 [50]	Brazil	RCT	1~30	1,3	16~31	88	132	No	AM
Gross 1998 [51]	Indonesia	CSS	30~114	3	20.2~30.6	81	81	Yes	AM
Gurgel 2018 [52]	Brazil	CSS	1–2	1	24.8~34.0	424	424	No	AM
Hampel 2017 [53]	Bangladesh	IS	60–120	3	18~22	17	17	Yes	AM
Haskell 2021 [54]	Malawi	RCT	180	3	19~31	103	103	No	NS
Jiang 2016 [55]	China	CS	1–42	1, 2, 3	20–35	102	306	No	AM
Khan 2007 [56]	Vietnam	RCT	174~342	4	21.2~31.2	268	268	NS	AM
Kim 1990 [58]	USA	CSS	30–210	3 + 4	NA	54	54	NS	AM
Kim 2017 [57]	South Korea	CSS	30–330	3, 4	28.6~34.8	334	334	Yes	random
Klevor 2016 [59]	Ghana	RCT	180	3	21.1~32.1	243	243	No	NS
Lira 2018 [60]	Brazil	CSS	2	1	18.3~35.5	134	134	No	AM
Liu 2016 [90]	China	CSS	3–180	1, 3	NA	43	43	NS	NS
Liyanage 2008 [61]	Sri Lanka	CSS	60–270	3 + 4	21.0~33.2	88	88	NS	NS
Lopez-Teros 2017 [62]	Mexico	CSS	30–150	3	22~32	56	56	No	AM
Machado 2019 [63]	Brazil	LS	85–105	3	20–40	19	19	No	AM
Martin 2010 [64]	Brazil	RCT	20–30	3	19.3~30.7	61	61	NS	AM
Matamoros 2018 [65]	Argentina	CSS	30–90	3	18~33	79	79	Yes	AM
Mello-Neto 2009 [66]	Brazil	CSS	20–60	3	16~44	136	136	NS	random
Meneses 2005 [67]	Brazil	CSS	28–83	3	20.3~32.9	49	49	Yes	AM
Muslimatun 2001 [68]	Indonesia	RCT	4–7	1	17~35	31	31	Yes	AM
Olafsdottir 2001 [69]	Iceland	CSS	60–120	3	27~35	77	77	No	NS
Ortega 1997 [70]	Spain	IS	13–40	2,3	24.2~31.6	57	114	No	AM
Palmer 2016 [71]	Zambia	RCT	120–360	3 + 4	18~30	140	140	Yes	AM
Palmer 2021 [72]	Zambia	RCT	270	4	21~34	216	216	Yes	AM
Panpanich 2002 [73]	Thailand	CSS	120–360	3, 4	19.2~31.6	226	226	No	NS
Redeuil 2021 [9]	Switzerland	CS	1–308	1, 3	27.0~35.4	49	102	Yes	AM
Ribeiro 2007 [74]	Brazil	CSS	1	1	18–40	24	24	No	NS
Rice 2000 [75]	Bangladesh	RCT	90	3	20.9~32.3	35	35	Yes	NS
Samano 2017 [76]	Mexico	CSS	30–60	3	19.0~35.0	32	32	Yes	AM
Sânzio Gurgel 2016 [77]	Brazil	CSS	1~7	1	24.6~32.6	25	25	No	AM
Schulz 2007 [78]	Germany	CSS	2	1	24.9~32.9	29	29	No	NS
Schweigert 2004 [79]	Germany	CSS	2–21	1, 3	24~36	21	42	Yes	NS
Souza 2015 [80]	Brazil	CSS	30	3	22.4~35.0	80	80	No	AM
Szlagatys-Sidorkiewicz 2012 [81]	Poland	LS	30–32	3	23.0~29.2	25	25	Yes	AM
Szlagatys-Sidorkiewicx 2012 [82]	Poland	CSS	3–32	1, 3	22.0~32.6	49	98	Yes	AM
Tijerina-Saenz 2009 [83]	Canada	CSS	30	3	20~40	60	60	No	NS
Tokusoglu 2008 [84]	Turkey	CSS	60–90	3	20–40	92	92	No	AM
Tomiya 2017 [85]	Brazil	RCT	1	1	18~31	101	101	NS	NS
Turner 2013 [86]	Bangladesh	RCT	78–267	3 + 4	20~26	135	135	Yes	NS
Vaisman 1985 [16]	Israel	CSS	7~28	1, 3	NA	7	14	Yes	random
Whitefield 2020 [87]	Cambodian	IS	21–187	3	21.4~30.7	68	68	Yes	NS
Zhang 2001 [89]	China	CSS	1–90	1, 2, 3	21~31	365	365	No	NS
Zhang 2021 [91]	China	CSS	1–330	1, 2, 3, 4	22.2~30.4	923	923	Yes	AM

Note: study design: CSS-Cross Sectional Study; CS- Cohort Study; IS-Intervention study; LS- Longitudinal study; RCT-Randomized Control Test. Lactating stage: 1 colostrum; 2 transitional human milk; 3 early mature human milk; 4 late mature human milk. * 3 + 4 Denotes there was no precise specification of mature human milk. ^&^ Multination included AU, CA, CL, CHN, JPN, MEX, PH, UK and USA. ^#^ AM: Morning. PM: afternoon or evening; NS: not specified; Random: single sampling and specified daytime but not clarify exact sampling hours.

**Table 2 nutrients-14-04844-t002:** Summary of findings for the comparison between Chinese and non-Chinese human milk samples.

Lactation Stage	Studies Enrolling Chinese Participants				Studies Enrolling Non-Chinese Participants				X^2^	*p*	Total Studies			
	No.	Sample Size	Mean	95% CI	No.	Sample Size	Mean	95% CI			No.	Sample Size	Mean	95% CI
Colostrum	5	429	1039.1	470.3, 1607.8	22	1741	895.8	714.1, 1077.4	0.22	0.64	27	2170	920.7	744.5, 1096.8
Transitional	3	356	505.7	118.0, 893.4	3	363	542.2	278.9, 805.6	0.02	0.88	6	719	523.7	313.7, 733.6
Mature	7	1268	386.4	270.6, 502.3	53	5014	385.2	335.1, 435.3	0.00	0.98	59	6282	385.4	339.4, 431.3
Early	7	1112	408.4	282.6, 534.1	38	3221	401.2	333.6, 468.6	0.01	0.92	44	4333	402.4	342.5, 462.3
Late	1	156	240.0	214.9, 265.1	5	794	259.3	220.8, 297.8	0.68	0.41	6	950	254.7	223.7, 285.7

## Data Availability

The data presented in this study are available in the inserted articles.

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
