# Peer review of "Vitamin A Concentration in Human Milk: A Meta-Analysis"

_nutrients, 2022, doi:10.3390/nu14224844_

Round 1

Reviewer 1 Report

This is a comprehensive and technically sound meta-analysis of a very large literature database which reports the concentrations of retinol (vitamin A) in human breast milk from different geographical regions around the world. The study design and the analytical procedures are appropriate for this type of investigation and the conclusions from the vast amount of data surveyed are sound.

The purpose of this study appears to have been to find the likely contribution of breast milk to the vitamin A status of infants under the age of 6 months, and whether there is a risk of vitamin A deficiency developing in some parts of the world where breast milk vitamin A concentrations could be quite low. One of the characteristics of vitamin A status in animals at various stages of life is that the liver is a very effective conservation and storage organ for vitamin A. There are powerful regulatory mechanisms that enable adequate concentrations of vitamin A in the circulation to be maintained by controlled release of vitamin A from the storage cells in liver. It would be helpful in the discussion of the findings from this meta-analysis if some comment or information could be provided about the significance of prenatal supply of vitamin A to the developing fetus from the maternal circulation. This really raises the question: is vitamin A status of newborn children determined by the dietary supply of vitamin A in milk or is this possibly a relatively minor source of vitamin A, compared to that which could have accumulated in the fetal liver during gestation? Some statement therefore in the Discussion, would be helpful, about the significance of vitamin A supply postnatally in milk, compared to the possible accumulation of vitamin A in the liver prenatally, on the maintenance of adequate vitamin A status in early postnatal life.

Because milk vitamin A concentration is probably related to the concentration in blood of the milk-producing mother, and because vitamin A concentration in blood is no indication of vitamin A reserves in the liver, as it is tightly regulated, Then, it is not surprising that milk vitamin A concentration in many milk samples is quite similar. Is the vitamin A status of the neonatal infant maintained by vitamin A in milk or is it really being maintained by hepatic store acquired prenatally?

Author Response

Point 1:This really raises the question: is vitamin A status of newborn children determined by the dietary supply of vitamin A in milk or is this possibly a relatively minor source of vitamin A, compared to that which could have accumulated in the fetal liver during gestation? Some statement therefore in the Discussion, would be helpful, about the significance of vitamin A supply postnatally in milk, compared to the possible accumulation of vitamin A in the liver prenatally, on the maintenance of adequate vitamin A status in early postnatal life.

Response 1: Two studies reported mean liver VA concentration of perinatal normal-weight newborns [Montreewasuwat N & Olson JA 1979, Olson JA et al 1984 ],one being 17.3±17.4 µg/g liver in Thai fetuses in gestational age of 37-40 week (n=10), the other being 22±26 µg/g liver in USA infants aged 0-6 days (n=22) . Assuming that the liver represents 4.3% of body weight and the liver VA concentration is 20 µg/g, a 3.2-kg full-term newborn has stores of 2.8 mg VA. In contrast, an exclusively breast-fed infant consumes approximately 54.3 mg of VA from mother’s milk (402.4µg/L×0.75l/day×180 days). About 19.4 times more VA is transferred from mother to infants during the 6 months of lactation than is accumulated by the fetus during 9 months of gestation. Obviously, the VA in breast milk is of paramount importance for maintaining adequate VA status in early postnatal life of infants as compared to accumulation of VA in the liver prenatally.

Reviewer 2 Report

Thank you for the opportunity to review this interesting paper. The goal of this review is to inform dietary reference intakes revision for the group of healthy full-term infants aged 0 to 6 months. The authors have been conducted a meta-analysis study to analyze human milk’s VA concentration on volume base determined by advanced methods (HPLC, UPLC, or UFLC). They explored theinfluence of potential confounders using meta-regression.

I have just some suggestion to improve this manuscript. I think that the introduction could be implemented expoloring other important functions of the VA. In particular is know the antiviral property of VA that could be mentioned. Recently a systematic review on this topic has been published "Sinopoli, A., Caminada, S., Isonne, C., Santoro, M. M., & Baccolini, V. (2022). What are the effects of vitamin A oral supplementation in the prevention and management of viral infections? A systematic review of randomized clinical trials. Nutrients14(19), 4081". Please added this reference. In the methodology I suggest to add a section dedicated to clinical trial. gov, a database of privately and publicy funded clinical studies conducted around the world. This would make the methodological research more complete.

Author Response

Point 1: I have just some suggestion to improve this manuscript. I think that the introduction could be implemented expoloring other important functions of the VA. In particular is know the antiviral property of VA that could be mentioned. Recently a systematic review on this topic has been published "Sinopoli, A., Caminada, S., Isonne, C., Santoro, M. M., & Baccolini, V. (2022). What are the effects of vitamin A oral supplementation in the prevention and management of viral infections? A systematic review of randomized clinical trials. Nutrients14(19), 4081". Please added this reference. 

Response 1: Latest research showed that VA may have protective effects on outcomes of some viral infections such as HPV and measles [Sinopoli et al 2022].  

Point2:  In the methodology I suggest to add a section dedicated to clinical trial. gov, a database of privately and publicy funded clinical studies conducted around the world. This would make the methodological research more complete.

Response 2: We feel pitiful that we didn’t apply for a registration of this meta-analysis on relevant meta-analysis registration platform prior to the performance and we will do so for this type of study next time.